# Abnormal Conductive State Identification of the Copper Rod in a Nickel Electrolysis Procedure Based on Infrared Image Features and Position Characteristics

**Rui Sun [1] , Gang Qin [1], Gaibian Li [2], Jinbao Hu [2], Jingqi Xiong [1] and Huanwei Xu [1,\*]**

[1] School of Mechanical and Electrical Engineering, University of Electronic Science and Technology of China, Chengdu 611731, China; rispper@uestc.edu.cn (R.S.); qingang6321@163.com (G.Q.); jqxiong@uestc.edu.cn (J.X.)

[2] JinChuan Group Co., Ltd., Jinchang 737100, China; ligaibian@dingtalk.com (G.L.); hjbzwy@dingtalk.com (J.H.)

\* Correspondence: hwxu@uestc.edu.cn

**Abstract:** In the nickel electrolysis industry, detection of the conductive state of copper rods is an important part of production procedure management, as abnormal conductive states of the copper rod result in a decline in the quality of the electrodeposited nickel plate. Conventional treatment consists of manual detection and handing, which is inefficient and induces more problems, such as the safety of the insulation. Because abnormal conductive states are only located between the copper rod and busbar, it has obvious position characteristics, and abnormal conductive states also induce a calorific difference in a particular area, which can be detected in an infrared image. We can use the infrared feature and position characteristics to identify the abnormal conductive faults. This paper introduces a method and practice for the identification of abnormal conductive faults in a conductive copper rod in the nickel electrolysis procedure using computer vision theory, including infrared image segmentation with position characteristics, infrared feature extraction, and conductive fault identification with SVM (support vector machine). The result shows that the method can divide the conductive states of the copper rod into abnormal heating conditions, normal operating conditions, and open circuit conditions, with a 90% accuracy rate on the obtained samples.

**Keywords:** abnormal conductive; state identification; infrared image; position characteristics

## 1. Introduction

In the nickel electrolysis industry, nickel products are mainly produced through the electrolytic refining process of the nickel sulfide anode in China. During the nickel electrolysis procedure, the running state of nickel electrolytic cells is closely related to the final quality of the cathode's nickel precipitation. The conductive state of the copper rod has a significant influence on the quality of the nickel plate. An abnormal conductive state of the copper rod is mainly caused by some factors, such as excessive local contact resistance, poor contact, and a short circuit, and it leads to differences in the surface temperature distribution, which are manifested as abnormal heating conditions and open circuit conditions. This exception can lead to a decline in the quality grade of the nickel plate. Thus, it is extremely important to rapidly detect and manage abnormal conductive states of the copper rods in the nickel electrolysis procedure.

Conventional manual inspection, which has low efficiency, high labor costs, and high labor intensity, is still widely used in the nickel electrolysis industry. Different conductive states lead to different surface temperature distributions of the copper rods, which can be detected and identified in an infrared image. A reasonable solution is to identify abnormal conductive states using infrared images and the AI (artificial intelligence) classification method. On this basis, non-contact measurement and automation can be applied in the nickel electrolysis industry

In the field of nickel electrolysis, Wang [1] studied and simulated the relationship between the current and surface temperature of a cathode copper rod, and the simulation results verified the feasibility of detecting the conductive state of copper rods based on infrared images. In addition to this case, less research and applications have focused on fault identification in copper rods based on infrared images. However, this method is commonly used in other fields, especially in the copper electrolysis field and electric power field, which has important value as a reference for solving the problem about the identification of the conductive states of copper rods in nickel electrolysis procedures.

In the field of copper electrolysis, Hu [2] proposed a Faster R-CNN (region-based convolutional neural network) convolutional neural network based on the focal loss function to identify short-circuit faults in copper rods. The authors of [3] proposed an automatic inspection system for the copper electrolysis procedure based on infrared images, which can identify short-circuit faults using SVM (support vector machine) with infrared image features, such as the mean gray, standard deviation, and Hu moment. The authors of [4] used the differential LBP (local binary pattern) method for infrared image feature extraction to reduce the influence of seasonal transformation and other factors, which improved the robustness of the classification model. The authors of [5] collected infrared images of a cathode copper bar of an electrolytic cell, obtained the surface temperature value, and solved the current value with COMSOL. In the field of electric power, Liang [6] proposed a method to identify the fault types of substation equipment using the relative temperature difference method based on the relationship between the equipment failure and temperature performance. The authors of [7] introduced methods for image preprocessing and feature extraction based on the infrared image of an insulator string, and realized the classification of insulator contamination grades using SVM. Then, Fu [8] used the decision tree model, which has the advantages of fast learning and intuitive and accurate classification, to identify deteriorated insulators. The authors of [9] introduced a transmission line insulator fault diagnosis method based on the binary support vector machine classifier and Bayesian optimization, which was used for the classification and recognition of the infrared spectrum in the process of insulator flashover. The authors of [10] selected six components of reciprocating compressors from an infrared image, and the average grayscale values were calculated to form 6-dimension vectors to represent the temperature distribution. Then, SVM was sued to diagnose the faults, with more than a 99% classification accuracy. According to the existing research works, state identification using infrared image features has mostly been used in long-distance monitoring or static detection while applications for near-field online detection are rare. Fault identification using infrared image features can reflect the temperature distribution and further identify the conductive states of copper rods, which is feasible for the detection of the conductive state of copper rods based on infrared images. Therefore, in this paper, this method is applied to identify the conductive state of copper rods with near-field online detection.

A nickel electrolysis workshop contains a large number of nickel electrolytic cells. Cathode and anode copper rods are placed alternately and equidistantly, which suspend cathode and anode nickel plate, respectively, by hangers. The diameters of cathode and anode copper rods are 35 and 45 mm, respectively. Both sides of the electrolytic cell have insulated partitions and busbars. In the traditional manual detection procedure, workers need to detect the conductive state of copper rods periodically by touching the copper rods and observing sparks in the instantaneous short circuit between cathode and anode copper rods. Figure 1 shows the nickel electrolytic cells and the conventional manual detection.

The conventional manual method has many disadvantages, such as low efficiency, heavy workload, high labor intensity, insulating safety, and missed detection.

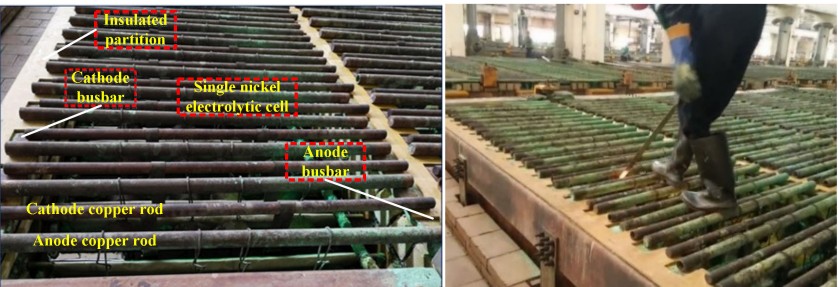

**Figure 1.** The nickel electrolytic cells and the conventional manual detection.

## 2. Solution Method

In nickel electrolysis, cells are supplied by 12V-DC power. DC currents start from anode busbars; pass through the anode hangers, anode plate, electrolyte, cathode plate, and cathode hangers; and finally reach the cathode busbars. Figure 2 shows the current distribution of conductive copper rods.

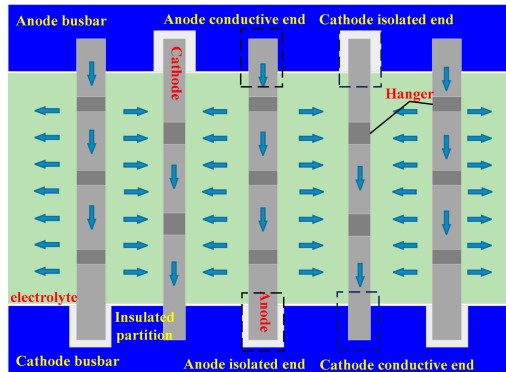

**Figure 2.** Current distribution of conductive copper rods.

We define the contact site between the copper rod and busbars as the "conductive end", and the contact site between the copper rod and the insulated partition as the "isolated end". Due to the current shunt of the anode hangers and the current converge of cathode hangers, the current gradually decreases to 0 in the copper rod from the conductive end to the isolated end. According to the simulation results of the relationship between the current and temperature in conductive copper rods, which was studied by Wang [1] and Zhao [3], from the conductive end to the isolated end, the temperature gradually decreases and approaches the environmental temperature. Figure 3 shows the temperature distribution of conductive copper rods.

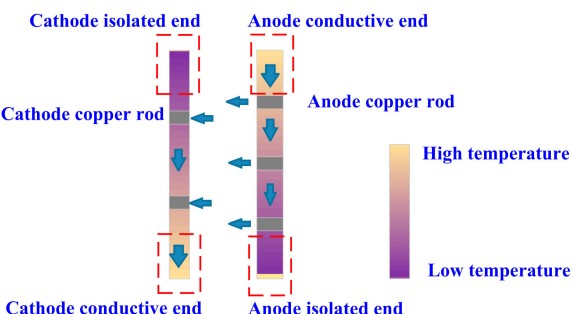

**Figure 3.** Temperature distribution of conductive copper rods.

In nickel electrolysis, there are three main kinds of conductive states of copper rods, namely the abnormal heating condition, normal operating condition, and open circuit condition. Different thermal effects caused by the actual current lead to different conductive

states, which lead to different surface temperatures of the copper rods. Figure 4 shows infrared images of copper rods in different conductive states.

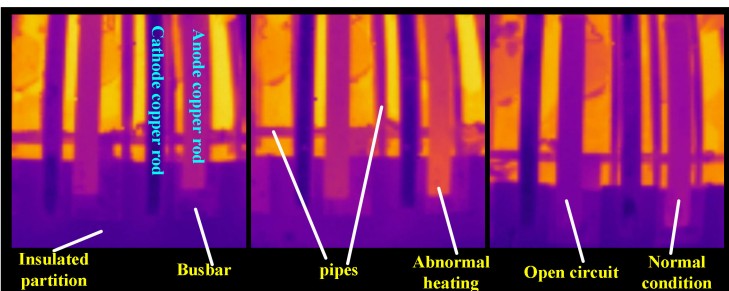

**Figure 4.** Infrared images of copper rods under different conductive states.

It can be seen that under the normal conductive state, the cathode rod presents a uniform brightness in the infrared image, which is brighter than the anode rod; a lower brightness in the open circuit state; and a higher brightness in the short circuit state. Due to the difference in the temperature of the background (electrolytic area), misjudgment of the direct temperature reaction of the conductive states may occur. It is more reasonable to judge the conductive states by comparing the differences between adjacent areas.

In Figure 4, all copper rods are placed vertically in the infrared image, and the anode rods are coarser than the cathode rods. As the background, the electrolytic zone consists of electrolytes and pipes in cells, the mean temperature of which is higher than the copper rods, busbars, and isolated partitions.

## 3. Infrared Image Segmentation

To reduce background interference, and ensure the effectiveness of infrared image feature extraction, it is necessary to segment copper rod areas from raw infrared images.

### 3.1. Copper Rod Segmentation Based on the Otsu Algorithm

In the raw infrared images, obvious brightness differences between the copper rods and electrolyte area exist, with obvious edges and contour features. Therefore, this paper attempts to use global threshold methods for infrared image segmentation. The Otsu algorithm is one of the most common global threshold methods and it searches for the best global threshold by traversing the gray histogram, which maximizes the variance between the background and foreground [11]. Figure 5 shows the results of the copper rod segmentation in different conductive states based on the Otsu algorithm.

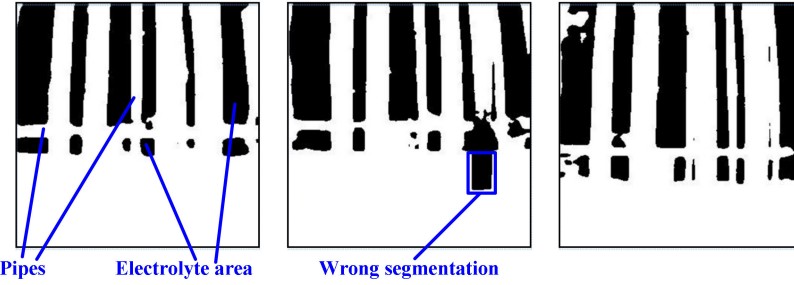

**Figure 5.** The results of copper rod segmentation based on the Otsu algorithm.

In Figure 5, some pipe areas are classified as foreground in the infrared image. Moreover, the copper rod under the abnormal heating condition is not segmented well because it is much brighter than the other copper rods and easily regarded as the electrolyte background.

However, the contour and edge of the cell wall area can be segmented well using the Otsu method. So, the approximate vertical position of the cell wall edge can be located

by searching for the position of the white-black jump points in a certain range from the bottom of the binary image [12,13]. Figure 6 shows the searching result.

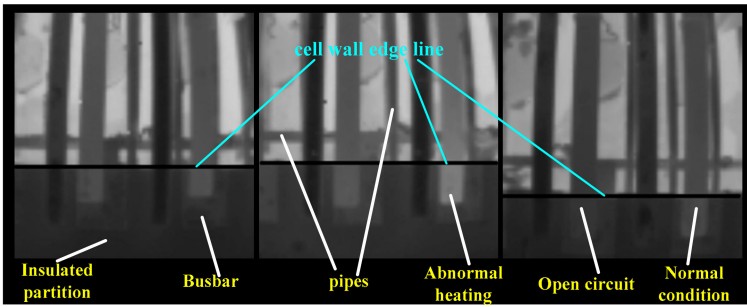

**Figure 6.** The approximate vertical position of the cell wall edge in different conductive states.

In Figure 6, the black lines represent the approximate vertical position of the cell wall edge in the grayscale image. It only covers the areas of the copper rods, insulated partitions, and busbars, and avoids the electrolyte areas being covered as much as possible. The areas of the conductive states that we are focused on are below the cell wall edge lines.

### 3.2. Copper Rod Segmentation Based on the Region Growing Algorithm

The global threshold method has certain limitations because of the gray properties of some areas, such as copper rods, pipes, and electrolytes, in grayscale images. The gray properties in each copper rod area are similar. We can use the local region growing algorithm to segment each of the copper rods in grayscale images.

#### 3.2.1. Selection of Initial Seeds

In the grayscale image, all copper rods with a fixed width are placed vertically and equidistantly, which are darker than the electrolyte area. The vertical gray mean shows differences in *x*-axis direction and the copper rod areas have the characteristics of a low gray mean. So, the initial seed selection for each copper rod can be determined by analyzing the mean, variance, and jump-edge of the pixel set in the copper rod area [14].

In this paper, we trim a $320 \times 100$ captured image from an original $320 \times 320$ grayscale image without isolated partitions and busbars. There are obvious differences between wide copper rod areas and electrolyte areas.

Figure 7 shows the trimming procedure and distribution of vertical gray mean along x axis.

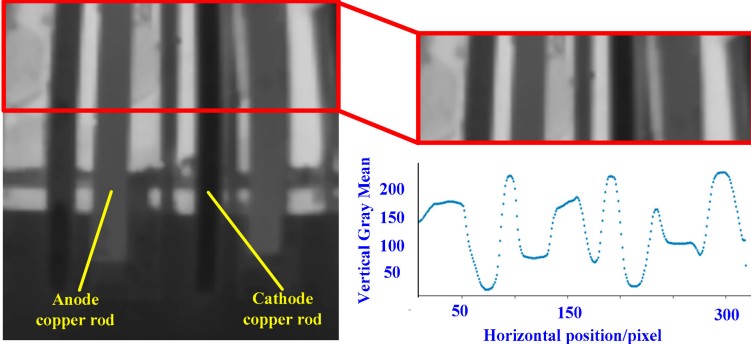

**Figure 7.** The original grayscale image, the captured image, and the scatter plot of the relationship between the horizontal position and vertical gray mean.

The initial seed selection for wide copper rods can be determined by traversing the vertical gray mean on the x axis and analyzing the mean, variance, and jump-edge in the scatter plot [15]. The algorithm steps used to search for the initial seed for the wide copper rod is described as follows:

(1) Traverse all horizontal positions in the captured image, calculate the vertical gray mean at each position, and store it in array named verticalMean.

(2) According to the width of the wide copper rod, set a variable named WinSlideSize for the sliding window length and initialize WinSlideSize to be equal to 20.

(3) Traverse array verticalMean, enter step (4), and start to search for the position of the wide copper rod in the valley when the traversal results meet the following conditions:

$$\begin{cases} verticalMean[i] \leq 130 \\ verticalMean[i] - verticalMean[i+1] > 5 \\ verticalMean[i+1] - verticalMean[i+2] > 5 \end{cases} \qquad (1)$$

(4) Traverse array *verticalMean* from the index *i*, and calculate the variance in the sliding window named *verticalMean* [j:j+WinSlideSize]. When the variance is less than 5 for the first time, record the position as the start. When the variance is greater than 5 for the first time, the position is recorded as the end.

(5) The positions of the initial seed for the wide copper rod are (start + end)/2, which are recorded in the array named WideRodPos. Then, return to step (3) and continue to traverse until all initial seeds of all wide copper rods are found.

Figure 8 shows an illustration of all initial seeds for the wide copper rod after applying the above algorithm. It also shows the initial seeds for narrow copper rods, which can be selected using the same method.

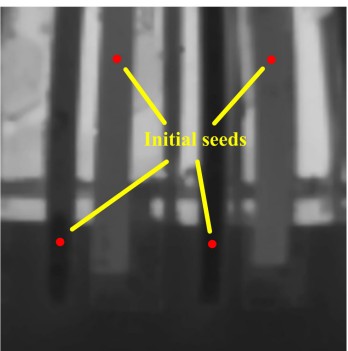

**Figure 8.** Illustration of all initial seeds.

### 3.2.2. Growing Criteria and Stopping Condition

The growth criterion of the region growing algorithm is a comparison rule used to determine the similarity between the seed and adjacent pixels or regions for extension to a larger region [16,17]. The stopping condition is a condition used for stopping regional growth if it does not match the growth criteria or exceeds the growth range [18,19]. In this paper, the gray difference discrimination method is used as the growth criterion and the limiting growth range condition is used as the stopping condition to segment each of the copper rods in the infrared image.

Gray Difference Discrimination Method

The gray difference discrimination method is used to calculate the absolute value of the gray value difference between the seed pixel and the neighborhood pixel. If it is less than a threshold T, the pixel is classified in the region where the seed is located for neighborhood extension at the seed position. Usually, four-neighborhood or eight-neighborhood expansion methods are used [20–22]. Figure 9 shows an illustration of the neighborhood expansion methods.

| Gray (i-1,j-1) | Gray (i-1,j) | Gray (i-1,j+1) |
|---|---|---|
| Gray (i,j-1) | Gray (i,j) | Gray (i,j+1) |
| Gray (i+1,j-1) | Gray (i+1,j) | Gray (i+1,j+1) |

**(a)**

| Gray (i-1,j-1) | Gray (i-1,j) | Gray (i-1,j+1) |
|---|---|---|
| Gray (i,j-1) | Gray (i,j) | Gray (i,j+1) |
| Gray (i+1,j-1) | Gray (i+1,j) | Gray (i+1,j+1) |

**(b)**

**Figure 9.** Illustration of the neighborhood expansion methods, (**a**) 4-neighborhood expansion, (**b**) 8-neighborhood expansion.

Limiting Growth Range Condition

Because, in infrared images, copper rods with a fixed width are placed vertically and equidistantly, the growth area can be limited to roughly cover a single copper rod width. Therefore, the position of pixel $I(i,j)$ in different growth areas should match the following conditions:

$$\begin{cases} 0 \leq i \leq EdgePos + 50 \\ x - 18 \leq j \leq x + 18 \end{cases} \qquad (2)$$

In Equation (2), $x$ is the position of the initial seed for the copper rod on the $x$-axis, and *EdgePos* is the position of the cell wall edge on the $y$-axis. Figure 10 shows an illustration of the limiting growth range condition.

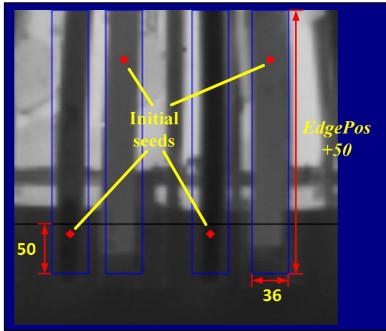

**Figure 10.** Illustration of the limiting growth range condition.

Figure 11 shows the result of the copper rod segmentation in different conductive states based on the region growing algorithm.

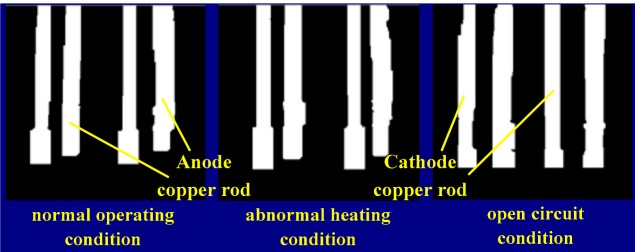

**Figure 11.** Results of the copper rod segmentation based on the region growing algorithm.

*3.3. Result of the Copper Rod Segmentation*

To segment the border of the copper rod and eliminate the interference in the electrolyte area as much as possible, the width of the boundary rectangle can be appropriately contracted. Figure 12 shows the result of the copper rod segmentation.

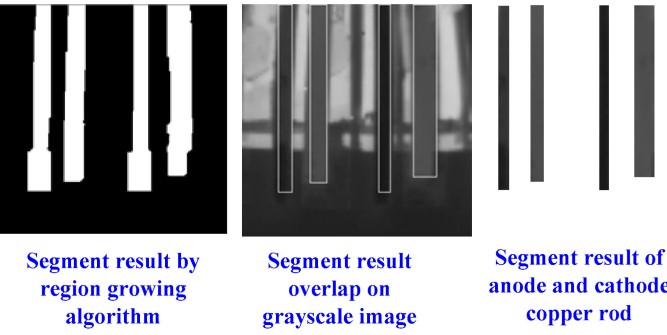

Segment result by region growing algorithm     Segment result overlap on grayscale image     Segment result of anode and cathode copper rod

**Figure 12.** Result of the copper rod segmentation.

## 4. Fault Identification

After infrared image segmentation, we can analyze and extract infrared image features, which are constructed as an infrared feature vector of the conductive copper rod. Then, a classification model is chosen to identify the abnormal conductive states.

### 4.1. Infrared Feature Extraction

We obtained 120 rod sample segmentations from the obtained samples through infrared image segmentation, which were regrouped in 40 sample sets. Each set includes three different samples and reflects three different conductive states.

In each set, brightness differences of the three sample segmentations exist, and uneven gray distributions and gray gradient distributions under the abnormal heating condition are observed. These different items can be described as follows:

(1)   Gray mean (*mean*)

The gray mean reflects the average temperature of the copper rod. The higher the gray mean is, the higher the temperature of the copper rod:

$$mean = \frac{1}{M \times N} \sum_{i}^{M} \sum_{j}^{N} I(i,j) \tag{3}$$

where *M* and *N* are the number of rows and columns of the copper rod sample gray image *I*.

(2)   Gray standard deviation (*std*)

The gray standard deviation reflects the discrete degree of the surface temperature distribution of the copper rod. The larger the value of the gray standard deviation, the higher the dispersion degree of the temperature:

$$std = \sqrt{\frac{1}{M \times N} \sum_{i}^{M} \sum_{j}^{N} [I(i,j) - mean]^2} \tag{4}$$

(3)   Mean gradient (*G*)

The mean gradient reflects the gradient variation of the copper rod surface temperature. From the conductive end to the isolated end, the temperature in the copper rod gradually decreases and approaches the environmental temperature. Gray gradient variation can be quantified in rod sample segmentations and this gradient variation is obvious under the abnormal heating condition but not clear under the other conditions. Figure 13 shows the gradient variation in different directions.

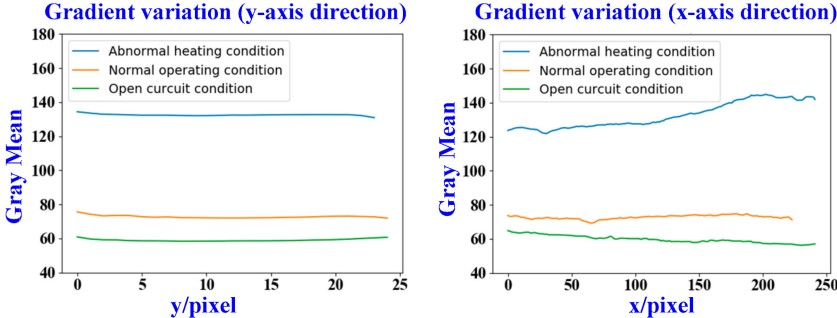

**Figure 13.** Gradient variation in different directions.

In Figure 13, the gradient variation in the *y*-axis direction is not obvious in all conductive states, but in the *x*-axis direction, gradient variation is more obvious under the abnormal heating condition than the others. So, the mean gradient of the sample segmentations in the *x*-axis direction can be used as an infrared image feature and is defined as follows:

$$G = \frac{1}{M \times N} \sum_{i}^{M} \sum_{j}^{N} \sqrt{[I(i+1,j) - I(i,j)]^2 + [I(i,j+1) - I(i,j)]^2} \tag{5}$$

To reduce the computation burden of the mean gradient, the result of the square root is approximately replaced by the absolute value:

$$G = \frac{1}{M \times N} \sum_{i}^{M} \sum_{j}^{N} |I(i+1,j) - I(i,j)| + |I(i,j+1) - I(i,j)| \tag{6}$$

(4)  Deformation of Hu moments ($H_1, H_2, H_3$ )

With the properties of translation, rotation, and scale invariant, the Hu moment is calculated and used as an image feature to identify copper rods under different conductive states [20]. For discrete two-dimensional images of $M \times N$, the $p + q$ order discretization origin moment is defined as follows:

$$m_{pq} = \sum_{x=1}^{M} \sum_{y=1}^{N} I(x,y) x^p y^q \tag{7}$$

To offset the influence of the change in the positon on the moment calculation in the target area and make it translation invariant, the $p + q$ order center moment is defined as follows:

$$\mu_{pq} = \sum_{x=1}^{M} \sum_{y=1}^{N} I(x,y)(x - x_c)^p (y - y_c)^q \tag{8}$$

In Equation (8), $(x_c, y_c)$ is the gray centroid coordinate of the image target, $x_c = m_{10}/m_{00}$, $y_c = m_{01}/m_{00}$.

The normalized center moments are defined as follows:

$$\eta_{pq} = \frac{\mu_{pq}}{\mu_{00}{}^r} \ , \quad r = \frac{p+q}{2} \tag{9}$$

On the basis of two-order and three-order normalized center moments, Hu moments can be defined as follows (the first three of seven) [22]:

$$\begin{cases} \Phi_1 = \eta_{20} + \eta_{02} \\ \Phi_2 = (\eta_{20} - \eta_{02})^2 + 4\eta_{11}{}^2 \\ \Phi_3 = (\eta_{20} - 3\eta_{12})^2 + 3(\eta_{21} - \eta_{03})^2 \end{cases} \tag{10}$$

Actually, conventional Hu moments include seven invariant moments, but after comparing all rod sample segmentations, it is found that only the first three Hu moments ($\Phi_1, \Phi_2, \Phi_3$) show obvious differences, which are used for the image feature in this paper. Hu moments always have a large range and can be negative values. The parameter $H_i$ is defined, which is the natural logarithm value of the absolute values of Hu moments:

$$H_i = \ln|\Phi_i| \quad i = 1, 2, 3 \tag{11}$$

(5) Gray difference between the target copper rod and the isolated end of the adjacent copper rod ($D$)

To reduce the influence of the environmental temperature on the identification results, the gray difference can be used to identify differences in the infrared feature between the target copper rod and the isolated end of the adjacent copper rod because the mean gray in the area of the isolated end represents the current environmental temperature. $D$ is defined as follows:

$$
\begin{aligned}
D &= mean1 - mean2 \\
mean1 &= \frac{1}{M_1 \times N_1} \sum_{i=1}^{M_1} \sum_{j=1}^{N_1} I_1(i,j) \\
mean2 &= \frac{1}{40 \times N_2} \sum_{i=M_2-40}^{M_2} \sum_{j=1}^{N_2} I_2(i,j)
\end{aligned}
\tag{12}
$$

where $mean1$ is the gray mean of the target copper rod area; $M_1$ and $N_1$ are the number of rows and columns; $mean2$ is the gray mean of the adjacent copper rod area, which is used for comparison; $N_1$ is the number of columns; and the number of rows is limited to 40.

Figure 14 shows an illustration of the calculation method for the infrared feature $D$.

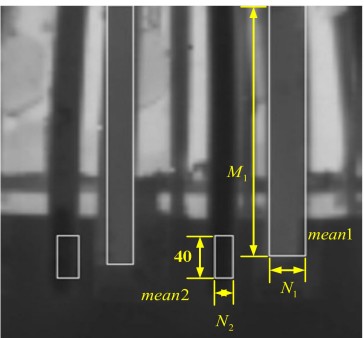

**Figure 14.** Illustration of the calculation method for the infrared feature $D$.

Table 1 shows the results of infrared feature extraction and the fault labels for part of the rod sample segmentations. The fault labels for the abnormal heating state, normal operation state, and short circuit state are set to 0, 1, and 2, respectively.

**Table 1.** Results of infrared feature extraction and the fault labels for the samples.

|  | *Mean* | *std* | *G* | $H_1$ | $H_2$ | $H_3$ | *D* | *Label* |
|---|---|---|---|---|---|---|---|---|
| 1 | 100.8 | 7.1 | 0.77 | 4.87 | 9.79 | 19.5 | 65.37 | 0 |
| 2 | 100.58 | 7.17 | 0.79 | 4.89 | 9.83 | 19.57 | 68.52 | 0 |
| 3 | 83.55 | 10.73 | 1.45 | 4.76 | 9.58 | 21.35 | 48.69 | 0 |
| 4 | 73.97 | 10.07 | 1.29 | 4.57 | 9.2 | 19.09 | 40.56 | 1 |
| 5 | 76.82 | 2.05 | 0.51 | 4.59 | 9.23 | 21.15 | 42.61 | 1 |
| 6 | 74.84 | 2.07 | 0.69 | 4.79 | 9.66 | 23.34 | 46.33 | 1 |
| 7 | 48.01 | 3.62 | 1.05 | 4.03 | 8.11 | 18.83 | 15.48 | 2 |
| 8 | 45.05 | 2.45 | 0.61 | 3.97 | 7.99 | 20.11 | 14.23 | 2 |
| 9 | 44.9 | 2.42 | 0.62 | 3.92 | 8.01 | 20.26 | 14.64 | 2 |

*4.2. States Identification Based on Support Vector Machine*

In Table 1, there are obvious differences in the infrared feature for the mean, std, and D, and setting a threshold seems to be an effective way to identify the conductive state types of the copper rod. However, when we apply this method to the obtained samples, the threshold identifying method cannot achieve an adequate accuracy. In general, SVM is more suitable for applications to limited samples and multidimensional image features [23,24].

SVM is a classical machine learning model with strict mathematical proof and strong interpretability, which has an excellent generalization ability on limited samples and has been widely used in infrared image recognition [25]. For fault identification in the conductive copper rod, SVM can reasonably predict and estimate the unknown dependency between the infrared feature input and fault label output, which tries to search for the best separating hyperplane in the infrared feature space.

This paper uses the soft margin support vector machine with RBF (radial basis function) to accomplish the fault identification task. SVM can tolerate some misclassification samples and avoid overfitting to a certain extent [26]. The original constraint problem for soft margin support vector machines is defined as follows:

$$\begin{cases} \min \frac{1}{2}\|w\|^2 + C\sum\limits_{i=1}^{m} \xi_i \\ s.t. \quad y_i(w^T x_i + b) \geq 1 - \xi_i, \quad \xi_i \geq 0, \quad i = 1, 2, \ldots, m \end{cases} \tag{13}$$

where $x$ is the input sample feature vector, $y$ is the classification label, $w$ is the weight distribution of SVM through iterative calculation, $\xi$ is the relaxation variable introduced for each sample point, $C$ is the adjustable penalty coefficient, and $a$ is the Lagrange coefficient.

The dual problem of soft margin support vector machines is defined as follows:

$$\begin{cases} \max \sum\limits_{i=1}^{m} a_i - \frac{1}{2}\sum\limits_{i=1}^{m}\sum\limits_{j=1}^{m} a_i a_j y_i y_j (x_i^T x_j) \\ s.t. \quad \sum\limits_{i=1}^{n} a_i y_i = 0, \quad 0 \leq a_i \leq C, \quad i = 1, 2, \ldots, m \end{cases} \tag{14}$$

As a kernel function, with the characteristics of less super-parameters, less calculation, and a strong learning ability, RBF can not only implicitly map sample data points to an infinite dimensional space but also calculate the product of high-dimensional feature vectors in a low-dimensional feature space. RBF is defined as follows:

$$\kappa(x_i, x_j) = \exp(-\gamma\|x_i - x_j\|^2), \quad \gamma = \frac{1}{2\sigma^2} > 0 \tag{15}$$

In the SVM classification model, the parameter $\gamma$ of RBF and the penalty coefficient $C$ of the soft interval SVM need to be adjusted. The parameter $\gamma$ affects the distribution of the sample data in the high-dimensional feature space. Moreover, the penalty coefficient $C$ is used to balance the size of the classification margin and accuracy of the model. To obtain an SVM model with a strong generalization ability, in this paper, the grid search method and k-fold cross validation method are used to identify the optimal super-parameter combination. Figure 15 shows the accuracy distribution map of the SVM model with the 5-fold cross validation method under different super parameter combinations. When $C^* = 1.4$ and $\gamma^* = 0.1$, the classification accuracy of SVM reaches 92.1%.

Finally, this paper selects the above optimal hyper-parameter combination to verify the performance of the SVM model for state identification using the obtained samples. Each state type contains 40 sample data, in which 30 are selected as training sets. The remaining samples form the test sets. After training, the classification accuracy of SVM reaches 90% using the test sets. Table 2 and Figure 16 show the classification results of the models using the test sets.

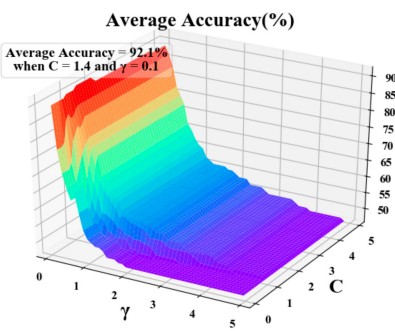

**Figure 15.** Accuracy distribution map of the SVM model under different super-parameter combinations.

**Table 2.** Statistical table of the classification results using the test sets.

| Conductive Conditions | Number of Test Samples | Correction | Misclassification | Accuracy | Total Accuracy |
|---|---|---|---|---|---|
| Abnormal heating condition | 10 | 7 | 3 | 70% | |
| Normal operation condition | 10 | 10 | 0 | 100% | 90% |
| Open circuit condition | 10 | 10 | 0 | 100% | |

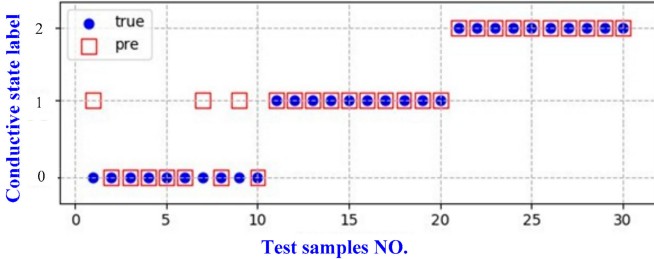

**Figure 16.** The classification results of the models using the test sets.

Only three copper rod samples under the abnormal heating condition are misclassified. Due to copper rust on the surface of the conductive copper rod and an insufficient infrared transmission effect, some sample images show irregular dark patches or texture, resulting in similar infrared feature distances. Figure 17 shows the comparison chart of one misclassified sample and the other two samples with the correct prediction.

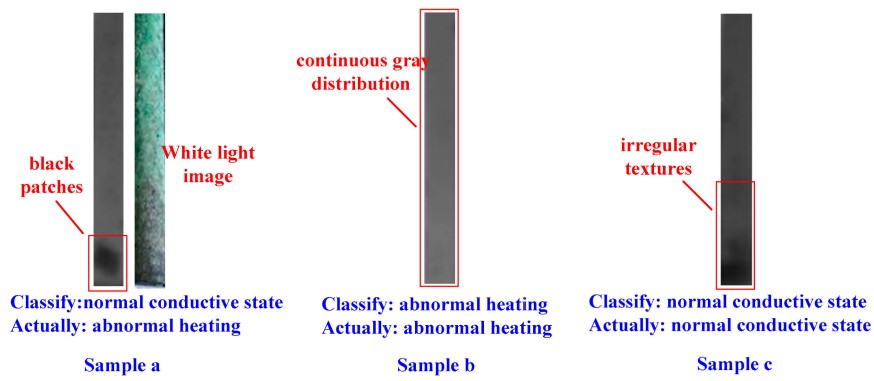

**Figure 17.** The comparison chart of the misclassified sample (**a**) and other samples (**b**,**c**) with the correct prediction.

Table 3 shows the results of the infrared feature extraction of these three samples. Further infrared feature distances are observed between sample a and b, but the infrared feature sample a is closer to that of sample c in the feature space, so the fault state of sample a can easily be misclassified as a normal operating condition.

**Table 3.** The results of infrared feature extraction of these three samples.

|   | *Mean* | *std* | *G* | $H_1$ | $H_2$ | $H_3$ | *D* |
|---|---|---|---|---|---|---|---|
| a | 83.55 | 10.73 | 1.45 | 4.76 | 9.58 | 21.35 | 48.69 |
| b | 100.58 | 7.17 | 0.79 | 4.89 | 9.83 | 19.57 | 68.52 |
| c | 73.97 | 10.07 | 1.29 | 4.57 | 9.2 | 19.09 | 40.56 |

At present, this method of conductive state identification can achieve a 90% accuracy on the obtained samples, so it has a certain practical significance for engineering applications. However, this method still has some defects and cannot identify individual samples well due to the close feature distance among these samples. For infrared image feature extraction, this paper mainly focuses on the global features of the sample while ignoring the local features. In the future, we can make full use of the obvious gradient change in the copper rod, and then introduce local features, such as the fluctuation amplitude, to increase the discrimination between samples. Moreover, to improve the generalization ability of the classification model, we need to expand the sample set and we can attempt to use models, such as the deep learning network, to achieve a higher accuracy.

**5. Conclusions**

This study used the infrared image features and position characteristics of the copper rod as much as possible, and realized image segmentation based on the local region growing algorithm. After infrared feature extraction, SVM was trained to identify the conductive state types, and the accuracy reached 90% on the obtained samples and matched the automatic detection requirements. However, this method still has some defects because the sample types are not comprehensive enough. Some misclassifications were identified due to the close feature distance among individual samples. In the future, to improve the generalization ability of the classification model, we need to expand the sample set and attempt to introduce local features, such as the fluctuation amplitude, to increase the discrimination between samples.

**Author Contributions:** Conceptualization, R.S.; methodology, G.Q.; software, H.X.; validation, R.S. and J.X.; formal analysis, G.L.; investigation, J.H.; resources, J.X.; data curation, J.H.; writing—original draft preparation, G.Q.; writing—review and editing, R.S.; visualization, G.Q.; supervision, J.X.; project administration, G.L. All authors have read and agreed to the published version of the manuscript.

**Funding:** This research received no external funding.

**Institutional Review Board Statement:** Not applicable.

**Informed Consent Statement:** Not applicable.

**Conflicts of Interest:** The authors declare no conflict of interest.

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
