# Peer review of "Abnormal Conductive State Identification of the Copper Rod in a Nickel Electrolysis Procedure Based on Infrared Image Features and Position Characteristics"

_applsci, doi:10.3390/app12073691_

Round 1
Reviewer 1 Report
In line 90, presumably it should be written “Fig. 2” instead of “Fig. 3”.
It seems that the international literature for this topic receives only slight consideration.
The figures are not always clear regarding the differences between the three conditions abnormal heating, open circuit and normal, especially for untrained eye. You have to search for the differences. It would be helpful to have an additional description in the text to indicate the distinguishing features, for example Fig. 4, Fig. 11 etc.
Author Response
In line 90, presumably it should be written “Fig. 2” instead of “Fig. 3”.
It has been modified in the manuscript.
It seems that the international literature for this topic receives only slight consideration.
Two references had been added in the manuscript.
The figures are not always clear regarding the differences between the three conditions abnormal heating, open circuit and normal, especially for untrained eye. You have to search for the differences. It would be helpful to have an additional description in the text to indicate the distinguishing features, for example Fig. 4, Fig. 11 etc.
More explanatory statements have been added before Figure 4.
Fig. 11 shows the window positions determined by the position Characteristics, which will be used to cut the gray image. Binary images in Figure 11 does not express any difference in different conduction states.
Reviewer 2 Report
The article “Abnormal Conductive State Identification of Copper Rod in Nickel Electrolysis Procedure Based on Infrared Image Features and Position Characteristics” is dealing with the IR image segmentation including position characterization, IR feature extraction and conductive faults identification (with SVM) for conductive state of copper rods identification.
My recommendation is major revision since there too many issues which require more information and explanation of statements, defining variables in equations and formulas and re-organisation of the manuscript. Additionally there are a lot of spelling and grammar errors. If authors are willing to make an in-depth revision, this manuscript can be reconsidered for publication.
General comments and questions.
- This manuscript has numerous spelling and grammar errors which should be corrected if the manuscript is re-submitted.
- Parameters in formulas should be defined and/or explained when first introduced.
- Acronyms and abbreviations should be explained when first introduced.
- References 1-9 are not corrected and consistently introduced in the manuscript. E.g. Wang 1 should be [9]. Pei 7 should be [9].
- A lot of careless assignments, statements and formulations are found in this manuscript which require in depth revision.
- Caption of Figures should be revised.
Detailed list of comments and questions:
Abstract
Line 12: “conductive state” instead of “Conductive state”
1. Introduction
Line 42: “It is reasonable” instead of “it is reasonable”
Line 43: AI classification. Abbreviation (artificial intelligence) should be explained when first introduced.
Line 45: “Wang 1” should be replaced by “[1]”
Line 52: “He 2” should be replaced by “[2]”
Line 52: Faster R-CNN. Acronym (Region-Based Convolutional Neural Network) should be explained when first introduced.
Line 53: “Zhao 3” should be replaced by “[3]”.
Line 56: “Jia 4” should be replaced by “[4]”.
Line 56: differential LPB method. Acronym (Local Binary Pattern) should be explained when first introduced.
Line 58: “Zhao 5” should be replaced by “[5]”.
Line 59: “obtained and solved” instead of “obtain and solve”.
Line 60: “Liang 6” should be replaced by “[6]”.
Line 62: “mastering the infrared image law of various equipment”: not clear to me. Please comment.
Line 62: “He 7” should be replaced by “[7]”.
Line 64: “Fu 8” should be replaced by “[8]”.
Line 66: “Pei 7” should be replaced by “[9]”.
Line 70: “monitoring” instead of “monitory”.
Line 86: “detection” instead of “detecting”.
2. Solution method
Line 90: “Fig.2” instead of “Fig.3”.
Line 93: “We define the contact site” instead of “We define that the contact site”.
Line 95: “what is current converge”? Please comment.
Line 97: “Wang 1 and Zhao 3” should be replaced by “[1] and “[3]”.
Line 99: “environmental temperature” instead of “environment temperature”.
Line 102: what is the difference between conductive state and conductive condition? Both terminologies are used throughout the manuscript. Please comment.
Line 111: “is” instead of “are”.
3. Infrared image segmentation
Line 133: “black lines” instead of “Black lines”.
Line 162: this line should be moved under line 163 (conditions for verticalMean) for a better understanding. Please comment.
Line 169: “Then return to step (3)” instead of “Then return step (3)“.
Line 171: “after applying above algorithm” instead of “after above algorithm”.
Lines 171, 174, 187 and 197: “illustration” instead of “sketch”.
Line 183: “is used to calculate” instead of “is to calculate”.
Lines 190: Caption of Figure 9: “Illustration of neighborhood expansion methods” instead of “shows the sketch of neighborhood expansion methods”.
Line 200: Caption of Figure 10: Illustration of limiting growth range condition” instead of “the sketch of limiting growth range condition.”
Line 200: Caption of Figure 11: “Results of copper rod segmentation based on region growing algorithm” instead of “the results of copper rod segmentation based on region growing algorithm.”
Line 209: “Segment result” instead of “Segmented resulte”.
Line 210: Caption of Figure 12: “Result of copper rod segmentation.” instead of “the result of copper rod segmentation.”
4. Fault identification
Line 217: “which can be regrouped” and “reflects” instead of “which can be regroup” and “reflect”.
Line 220: “uneven gray distributions and gray gradient distributions” instead of “uneven gray distribution and gray gradient distribution”.
Line 225: M and N are not defined when introduced in equation (3). Please comment.
Line 234: “environmental temperature“ instead of “environment. temperature“
Lines 239-240: This statement is in conflict with Figure 13. Please comment.
Lines 250-252: mpq and µpq are not defined eq. (7) and eq. (8) respectively. Please comment.
Lines 247-260: Definition of Hu moments is incomprehensible: H1, H2, H3 or F1 F2 F3? Please comment.
Line 267: “on identification results” instead of “to identification results”.
Line 268: “can be used for differences in infrared features between” instead of “can be used for infrared feature between”.
Line 269: “environmental temperature” instead of “environment temperature”.
Line 270: “D is defined” instead of “D are defined”.
Lines 271: N1 and N2 are not defined when first introduced. Please comment.
Line 272: “illustration” instead of “sketch”.
Line 275: Caption of Figure 14: “Illustration of calculation method for infrared feature D” instead of “The sketch of calculation method for infrared feature D.”
Lines 276, 279 and 281: “Table 1” instead of “Table.1”.
Line 279:”Results of infrared feature extraction and the fault labels for samples” instead of “the results of infrared feature extraction and the fault labels for samples”.
Line 282: “setting threshold seems to be” instead of “setting threshold seem to be”.
Lines 294-297: Variables should be properly defined when introducing equations (13) and (14). Please comment.
Line 304: “need” instead of “needs”.
Line 312: Caption of Figure 15: “Accuracy distribution map of SVM model under different super-parameter combinations” instead of “the accuracy distribution map of SVM model under different super-parameter combinations”.
Lines 316 and 318: “Table 2” instead of “Table.2”.
Line 322: “appear as irregular dark patches” instead of “appear irregular dark patches”.
Line 328 and 332: “above samples?? Should be specified. Please comment.
Lines 328 and 332: “Table 3” instead of “Table.3”.
Lines 328-331: samples a, b and c are not assigned in Figure 17, nor in caption of Figure 17. Please comment.
Author Response
- All abbreviations in this manuscript have been given the expression when first introduced.
- The quotes in the manuscript have been modified according to the comments of reviewers.
Figure captions and expression errors pointed out by reviewers have been modified in the manuscript.
- Line 62: “mastering the infrared image law of various equipment”: not clear to me. Please comment.
What we want to express here is that we have learned the relationship and law between equipment failure and temperature performance, and the sentence in manuscript has been modified.
- Line 95: “what is current converge”? Please comment.
The current of the cathode copper rod is transmitted to the cathode plate through 2-3 lugs, so the current density on the whole copper bar is unevenly distributed along the axial direction.
- Line 102: what is the difference between conductive state and conductive condition? Both terminologies are used throughout the manuscript. Please comment.
That’s problem of negligence, “conductive condition” has been replaced by ”conductive state” in the manuscript.
- Line 162: this line should be moved under line 163 (conditions for verticalMean) for a better understanding. Please comment.
For better understanding, the sentences order has been modified.
- Line 225: M and N are not defined when introduced in equation (3). Please comment.
The definition of M and N has been added in the manuscript
- Lines 239-240: This statement is in conflict with Figure 13. Please comment.
The description of sentence is correct, the annotations in the figure are wrong, which are not consistent with the horizontal labels, and Figure.13 has been modified.
- Lines 250-252: mpq and µpq are not defined eq. (7) and eq. (8) respectively. Please comment.
The definition of these to parameters has been added before eq. (7) and eq. (8).
- Lines 247-260: Definition of Hu moments is incomprehensible: H1, H2, H3 or F1 F2 F3? Please comment.
Hu moments in equation (10) are the original form, but we only use the first three of seven, then parameters Hi are practical form, which have been used and accepted by researchers and engineers, the relationship between Hu moments and parameters Hi is described in the manuscript.
- Lines 271: N1 and N2 are not defined when first introduced. Please comment.
N1 is the columns number of target copper rod area, N1 is the columns number of adjacent copper rod area. It has been modified in the manuscript.
- Lines 294-297: Variables should be properly defined when introducing equations (13) and (14). Please comment.
The definitions of x, y, w, a, C and ξ have been added after eq. 13.
- samples a, b and c are not assigned in Figure 17, nor in caption of Figure 17. Please comment.
Label a, b and c have been added in Figure 17, the caption of Figure 17 and references in manuscript have been modified.
Round 2
Reviewer 2 Report
I have read the revised version of the article “Abnormal Conductive State Identification of Copper Rod in Nickel Electrolysis Procedure Based on Infrared Image Features and Position Characteristics”
Authors have answered the comments and questions of the reviewer in a detailed way and have implemented changes and additional information where needed.
I am convinced that the overall quality of the article has improved considerably. Therefore I support and recommend publication of the revised manuscript after minor revision.
Minor comments
Line 86: “Fig.1 shows” instead of “Fig.1 is”
Line 169: “horizontal positions” instead of “horizontal position”
Line 211” illustration” instead of “sketch”
Line 256: “as an infrared image feature and is defined as follow” instead of “as a infrared image features and it is defined as follow”
Line 307: “In general“ instead of “In general case”
Line 320: “ξ is the relaxation variable” instead of “ξ is relaxation variables”
Line 342: “Table 2” instead of “Table.2”
Author Response
I have revised the manuscript according to the comments. Thank you for your hard work.
